# High Seroprevalence of SARS-CoV-2 in Mwanza, Northwestern Tanzania: A Population-Based Survey

**DOI:** 10.3390/ijerph191811664

**Published:** 2022-09-16

**Authors:** Helmut A. Nyawale, Nyambura Moremi, Mohamed Mohamed, Johnson Njwalila, Vitus Silago, Manuel Krone, Eveline T. Konje, Mariam M. Mirambo, Stephen E. Mshana

**Affiliations:** 1Department of Microbiology and Immunology, Weill Bugando School of Medicine, Catholic University of Health and Allied Sciences, Mwanza P.O. Box 1464, Tanzania; 2National Public Health Laboratory, Dar es Salaam P.O. Box 9083, Tanzania; 3East Central and Southern Africa Health Community, Arusha P.O. Box 1009, Tanzania; 4Infection Control and Antimicrobial Sterwardship Unit, University Hospital Wuerzburg, 97080 Wüerzburg, Germany; 5Department of Epidemiology and Biostatistics, School of Public Health, Catholoic University of Health and Allied Sciences, Mwanza P.O. Box 1464, Tanzania

**Keywords:** SARS-CoV-2, COVID-19, seroprevalence, antibodies, Mwanza, Tanzania

## Abstract

The transmission of the SARS-CoV-2 virus, which causes COVID-19, has been documented worldwide. However, the evidence of the extent to which transmission has occurred in different countries is still to be established. Understanding the magnitude and distribution of SARS-CoV-2 through seroprevalence studies is important in designing control and preventive strategies in communities. This study investigated the seropositivity of the SARS-CoV-2 virus antibodies in the communities of three different districts in the Mwanza region, Tanzania. A household cross-sectional survey was conducted in September 2021 using the modified African Centre for Disease and Prevention (ACDC) survey protocol. A blood sample was obtained from one member of each of the selected households who consented to take part in the survey. Immunochromatographic rapid test kits were used to detect IgM and IgG SARS-CoV-2 antibodies, followed by descriptive data analysis. Overall, 805 participants were enrolled in the study with a median age of 35 (interquartile range (IQR):27–47) years. The overall SARS-CoV-2 seropositivity was 50.4% (95%CI: 46.9–53.8%). The IgG and IgM seropositivity of the SARS-CoV-2 antibodies was 49.3% and 7.2%, respectively, with 6.1% being both IgG and IgM seropositive. A history of runny nose (aOR: 1.84, 95%CI: 1.03–3.5, *p* = 0.036), loss of taste (aOR: 1.84, 95%CI: 1.12–4.48, *p* = 0.023), and living in Ukerewe (aOR: 3.55, 95%CI: 1.68–7.47, *p* = 0.001) and Magu (aOR: 2.89, 95%CI: 1.34–6.25, *p*= 0.007) were all independently associated with SARS-CoV-2 IgM seropositivity. Out of the studied factors, living in the Ukerewe district was independently associated with IgG seropositivity (aOR 1.29, CI 1.08–1.54, *p* = 0.004). Twenty months after the first case of COVID-19 in Tanzania, about half of the studied population in Mwanza was seropositive for SARS-CoV-2.

## 1. Introduction

The coronavirus disease of 2019 (COVID-19), caused by the severe acute respiratory syndrome coronavirus-2 (SARS-CoV-2), still poses a major threat in healthcare facilities and is regarded as the most important public health problem worldwide. Cases of COVID-19 have been reported by 55 African member states and the local transmission has been well documented in most places [1,2]. Africa accounts for 17% of the global population; however, by late May 2021, it accounted for only 2.82% of the global COVID-19 cases and 3.7% of the global COVID-19 deaths reported [3]. This disparity has been attributed to a limited capacity for diagnosis, timely implementation of stringent containment measures, a younger population structure, and a predominance of asymptomatic and mild infections [4]. Since the index case of COVID-19 in Tanzania on 16 March 2020 [5], there have been no seroprevalence studies in the region of Mwanza to estimate the magnitude of the SARS-CoV-2 infections.

In low- and middle-income countries (LMICs), such as Tanzania, important public health measures, including social distancing or a lockdown, are difficult to implement owing to socioeconomic constraints. The testing capacity has increased substantially over a short period in different countries, but taking into considerations that most cases are asymptomatic or with mild symptoms, the actual spread of infections has not been able to be established [6].

Understanding the extent of the SARS-CoV-2 spread is particularly important for guiding COVID-19 mitigation efforts in light of the situation in Tanzania, whereby the handling of the COVID-19 pandemic took different turns [7]. The seroprevalence of SARS-CoV-2 worldwide differs from one country to another and among different populations in the same country [8]. However, the seroprevalence of SARS-CoV-2 in Africa has been found to range from 19.7% to 26% [9,10] in sub-Saharan Africa.

Different factors, such as limited access to healthcare and COVID-19 testing, limited surveillance, and lack of knowledge about SARS-CoV-2 infections amongst the population, have been reported to mediate the spread of SARS-CoV-2 [7]. However, these are not isolated challenges for Africa and may have a major role in the way SARS-CoV-2 spreads and behaves [11]. SARS-CoV-2 serosurvey data are critical for planning effective mitigation strategies and understanding the actual spread of the virus causing COVID-19 [12].

This study is aimed at looking at the serological prevalence of SARS-CoV-2 using the Africa Center for Disease Control and Prevention generic study protocol, which was meant to assist the African Union Member States in conducting standardized serological prevalence surveys. The information gained from these surveys will provide critical insight into the transmission and impact of the virus in Africa.

## 2. Materials and Methods

### 2.1. Study Design and Target Population

This was a cross-sectional, population-based serosurvey involving household members aged 5 years or above, regardless of their previous or current infection with COVID-19, who resided in the Magu, Misungwi, and Ukerewe districts in the Mwanza region during the period of transmission of SARS-CoV-2.

The estimated total population of Mwanza is about 3,122,992 (National Bureau of Statistics, 2016).

### 2.2. Study Area

The study was performed in Mwanza, Tanzania. The Mwanza region comprises of 8 district councils, whereby 7 of them are on the mainland. The study was performed in 3 out of the 8 district councils of the Mwanza region, Tanzania. The selection of the districts was based on the geography of the region and the administrative level of the specific districts. The aim was to include both Mwanza, mainland and island, thus, we selected two districts that are within mainland Mwanza, specifically Misungwi and Magu, and the Ukerewe Island. The Misungwi district was selected because it could be reached not only through the roads but also the railway (Central Railway Line), while Magu can only be reached via the roads. Ukerewe can only be reached via the waterway. Misungwi being a semi-urban district while Magu being a rural was another reason for the selection in order to have a presentation for both semi-urban and rural populations in the region.

### 2.3. Sample Size Estimation and Sampling Technique

The Kish Leslie formula was used to estimate the sample size using a prevalence of 22.3% from a previous study performed in Juba, Southern Sudan [12]. The estimated minimum sample size was 267. However, a total of 805 participants were enrolled. Using a randomized multi-stage cluster sampling technique, 3 districts were conveniently selected. The 3 ward administrative units were randomly selected from each of the selected districts. In each of the selected administrative wards, 3 villages were selected, thus, making 9 villages in total. Within each selected village, 10 communities were randomly selected using simple random sampling. In each community, 10 households underwent a random selection from the number of households provided. A member from the selected household of the age of 5 years and above was randomly selected to take part in a survey, which was expected to enroll 900 participants; however, 805 consented to participate in the study. A group of people (2 or more) living in the same compound was [13] regarded as a household.

### 2.4. Eligibility Criteria

#### 2.4.1. Inclusion Criteria

All household members aged 5 years and above were recruited for the investigation without taking into consideration their prior, recent, or current SARS-CoV-2 infection or current illness. The enrollments involved only the household members who have been residing in the study area during the period between March 2020 (the first case of SARS-CoV-2 was reported in March 2020 in Tanzania) and the date of enrollment and were able to give informed consent for those above the age of 18 and above. For children under the age of 18 years, their legal guardians were requested to give consent in the presence of a third party (witnesses).

#### 2.4.2. Exclusion Criteria

Household members who received any of the COVID-19 vaccines were excluded from the study.

### 2.5. Sample Collection

About 5 milliliters of blood were drawn by venipuncture from each of the selected members in each household once they consented to take part in the study. The blood was drawn by a trained phlebotomist, and it was collected in plain vacutainer tubes. The collected whole blood in the vacutainer tube was centrifuged at 2000 revolutions per minute for 15 min to obtain the sera. The sera were stored at −80 °C until they were processed. 

### 2.6. Serological Testing

The tested sera were screened for both IgM and IgG SARS-CoV-2 specific antibodies using a RADI COVID-19 IgG/IgM rapid test kit (KH Medics, Limited, Gyeonggi-do, Republic of Korea) with a sensitivity and specificity of 90% and 100%, respectively. All tests were performed following the guidelines written by the manufacturer; the quality control (QC) directives were followed accordingly. The test results were interpreted according to the manufacturer’s instructions. The SOPs during the pre-analytical, analytical, and post-analytical were observed throughout the study period. Known standard panels were part of the test assessments (Appendix A).

### 2.7. Survey Data Collection

Social demographic data and data regarding previous history of signs and symptoms suggestive of SARS-CoV-2 infections or COVID-19 were collected using a standardized data collection tool.

### 2.8. Data Analysis

A descriptive statistical analysis was conducted using STATA, version 12 (StataCorp LLC 4905 Lakeway Drive, College Station, TX, USA)

USA. Proportions were calculated for the categorical variables, and the median with an interquartile range was analyzed for the continuous data. A chi-square was used to test for the association between the categorical variables and the outcomes. Univariate and multivariate regression analyses were performed to assess the factors associated with the IgG and IgM seropositivity. The analysis of the factors associated with the IgG and IgM seropositivity was performed using bivariate analysis; factors with a *p* value of less than 0.5 were further analyzed in multivariate analysis, and variables with a *p* value of less than 0.05 were considered statistically significant.

### 2.9. Infection Prevention Measures against SARS-CoV-2 Infection Taken during Data Collection

All members of the research team involved had training in infection prevention and control measures. All research assistants were required to observe the hand hygiene between participants, the wearing of surgical masks, and observe the social distancing during the data collection.

## 3. Results

A total of 805 participants took part in the survey. The median age of the participants was 35 (IQR: 26–47) years. More than half of the participants, 56.9% (458), were males (see Table 1).

### 3.1. Previous History of Signs and Symptoms of COVID-19

Of the clinical signs and symptoms assessed, more than two thirds of the participants, 68.7% (553), reported no history of fever. The majority, 88.4% (712), reported no history of sore throat (see Table 2).

### 3.2. History of Chronic Illness

The majority of the participants had no history of chronic diseases or conditions, such as diabetes mellitus, 96.3% (775), sickle cell anemia, 99.4% (800), and hypertension, 95.5% (769). Almost all of the participants had no history of renal diseases, 99% (798), and cancer, 99% (797) (see Table 3).

### 3.3. Seroprevalence of SARS-CoV-2 Antibodies

The overall SARS-CoV-2 seropositivity was 50.4% (95%CI: 46.9–53.8) (406/805). The IgG and IgM seropositivity of the SARS-CoV-2 antibodies was 397 (49.3%) and 58 (7.2%), respectively, with 49 (6.1%) being both IgG and IgM seropositive, as shown in Figure 1.

### 3.4. Geographical Distribution of the Participants Who Tested for IgG against SARS-CoV-2

The clustering of the IgG positivity was observed in the two mainland districts of Magu and Misungwi, with similar observations in the island district of Ukerewe (Figure 2).

### 3.5. Factors Associated with IgG Seropositivity among the Participants

According to the bivariate analysis, only living in the Ukerewe district was associated with IgG seropositivity among the participants. According to the multivariate analysis, living in Ukerewe was independently associated with IgG seropositivity (adjusted odds ratio (aOR) 1.29, 95% confidence interval (CI) I 1.08–1.54, *p* value 0.004) (see Table 4).

### 3.6. Factors Associated with SARS-CoV-2 IgM Seropositivity among the Participants

According to the bivariate analysis, sex, district, history of runny nose, and history of loss of taste were significantly associated with IgM seropositivity. All remained significant in the multivariate logistic regression analysis as follows: history of runny nose (aOR 1.84, 95%CI 1.03–3.5 and *p*-value 0.036), history of loss of taste (aOR 1.84, 95%CI 1.12–4.48 and *p*-value 0.023), Magu district (aOR 2.89, 95%CI 1.34–6.25, *p*-value 0.007), and Ukerewe district (aOR 3.55, 95%CI 1.68–7.47, *p*-value 0.001) were independently associated with SARS-CoV-2 IgM seropositivity among the participants who took part in the study (see Table 5).

## 4. Discussion

The seroprevalence survey is an important method for estimating the actual burden of a disease. The specific IgM and IgG markers in the blood are key in establishing the extent of an infectious disease. The markers can indicate recent and past infections, showing the real magnitude of a disease in a given population. In this population-based sero-surveillance study, it was observed that in Mwanza, Tanzania about half of the population studied was SARS-CoV-2 seropositive. This seroprevalence was observed 20 months after the first case of COVID-19 was announced in Tanzania.

None of the study participants had received a COVID-19 vaccine, indicating that the presence of anti-spike protein antibodies was due to a previous infection. The findings in the current study, for which the seropositivity was 50.4% with an IgG seroprevalence of 49.3%, are consistent with the study performed in Gauteng, South Africa, which reported an IgG seroprevalence of 56.2% [14], and the recent results from Zanzibar, Tanzania, which reported a seroprevalence of 58.9% before the omicron wave [15]. However, studies that were performed among the East African countries of Kenya, the Democratic Republic of Congo, and South Sudan reported a low seroprevalence, ranging from 18.6–26.7% [12,16,17]. This could be due to the fact that these studies were performed during the first and second COVID-19 waves, while the current study was performed in the third wave. Furthermore, compared to other East African countries, Tanzania did not take stringent measures to control the spread of SARS-CoV-2, such as a complete lockdown. 

The IgM seropositivity was found to be relatively low, at 7.2%, indicating that less than 10% of the participants had recent infections. Similar studies in other African countries reported almost a similar level of IgM seropositivity, which was between 6.3% and 11.6% Slight deviations have been observed depending on the variations in the SARS-CoV-2 waves at which the studies were performed [17,18].

This study observed that a recent history of runny nose, a history of loss of taste, and living in the Misungwi and Ukerewe districts were independently associated with SARS-CoV-2 IgM seropositivity among the participants. A number of studies around the world reported loss of taste and smell to be the most common presentation of COVID-19 [19,20]. It is more likely that these participants were recovering from SARS-CoV-2 infections [21,22]. A history of runny nose also independently predicted the IgM seropositivity among the participants who took part in the survey [23].

This study also showed that living in the Ukerewe district was associated with IgG seropositivity. This could be explained by geographical variations since the Ukerewe district is an island, which is usually accessed through the lake, with limited air access. Being an island could lead to high interactions with limited movements within the island with many fishing activities. Furthermore, Nansio, the port town in Ukerewe, serves as the single social and financial hub of the district, with very high interactions among the people in this hub. Thus, people from all over the island come to Nansio to carter to their social and financial needs. Additionally, being a port town makes it more likely to encounter the spread of infections from the mainland districts through the visitors coming to Ukerewe. On the contrary, Magu and Misungwi are mainland districts with easy accessibility to social and economic activities available in the city of Mwanza. In addition, the results are supported by the observed high seroprevalence of 58.9% in the Zanzibar Island, Tanzania before the omicron wave [15]. The IgG results are further supported by the IgM results, showing that the odds of being IgM positive were significantly higher for Ukerewe than for Magu and Misungwi.

### Limitation

A limitation of the study is the possibility of cross reactivity of the antibodies of SARS-CoV-2 with other viruses with cross-antigenic similarities. However, based on the manufacturer’s details, the test was found to have 100% specificity, therefore, cross reactivity was unlikely. Due to the fact that the study was conducted in three districts that were conveniently selected, the study might not reflect the levels of IgG and IgM in the population in the Mwanza region. 

## 5. Conclusions

Twenty months after the first case of COVID-19 in Tanzania, about half of the studied population in Mwanza was seropositive for SARS-CoV-2. The IgM seropositivity was associated with a recent history of a runny nose and loss of taste. There is a need for continuing seroprevalence studies to estimate the magnitude and trends of SARS-CoV-2 infections in different populations in Tanzania.

## Figures and Tables

**Figure 1 ijerph-19-11664-f001:**
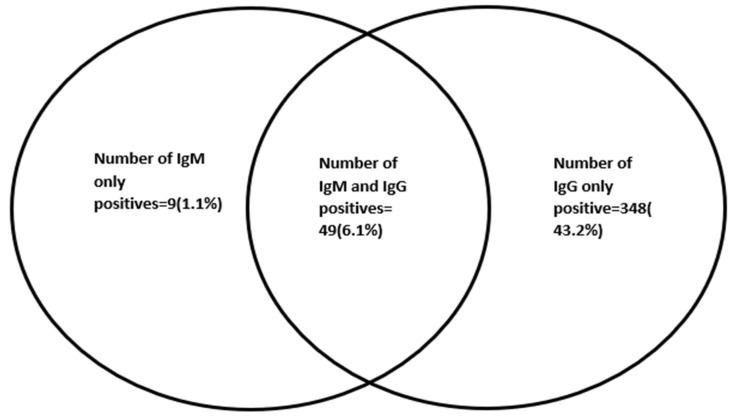
Venn diagram showing the distribution of serological markers among the participants.

**Figure 2 ijerph-19-11664-f002:**
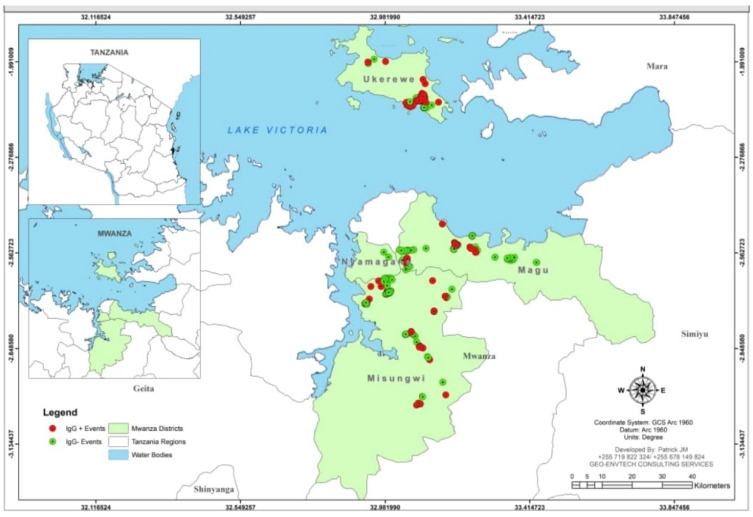
Geographical distribution of the participants with SARS-CoV-2 IgG positive in the districts of Misungwi, Magu, and Ukerewe in Mwanza, Tanzania.

**Table 1 ijerph-19-11664-t001:** Basic demographic data of the participants.

Variable	Frequencies (n)	Percentages (%)/Median/IQR
Age(years)	805	35 (IQR: 26–47)
**Sex**		
Male	458	56.9
Female	347	43.1
**District of residence**		
Magu	249	30.9
Misungwi	297	36.9
Ukerewe	259	32.2
**Occupation**		
Business	145	18.0
Employed	56	07.0
Farmers	448	55.7
* Others	156	19.3

* Housewives and children under the age of 18 years.

**Table 2 ijerph-19-11664-t002:** Previous history of signs and symptoms of SARS-CoV-2 infections.

Variables	Frequencies	Percentages (%)
History of fever		
Yes	252	31.3
No	553	68.7
History of fatigue		
Yes	265	32.9
No	540	67.1
History of sore throat		
Yes	93	11.6
No	712	88.4
History of cough		
Yes	585	27.3
No	220	72.7
History of runny nose		
Yes	275	34.2
No	530	65.8
History of shortness of breath(SOB)		
Yes	52	6.5
No	753	93.5
History of wheezing		
Yes	49	6.1
No	756	93.9
History of chest pain		
Yes	32	4.0
No	773	96.0
History of other respiratory symptoms		
Yes	11	1.4
No	794	98.6
History of headache		
Yes	259	32.2
No	546	67.78
History of smell loss		
Yes	64	7.9
No	741	92.1
History of taste loss		
Yes	82	10.2
No	723	88.8
History of diarrhea		
Yes	71	8.8
No	734	91.2

**Table 3 ijerph-19-11664-t003:** History of chronic illnesses or conditions among the participants.

Variable	Frequency (n)	Percentage (%)
History of diabetes		
Yes	30	3.7
No	775	96.3
History of lung diseases		
Yes	19	2.4
No	786	97.6
History of blood pressure		
Yes	36	4.5
No	769	95.5
History of renal diseases		
Yes	7	1
No	798	99
History HIV/AIDs		
Yes	20	2.5
No	785	97.5
On ART		
Yes	16	80
No	4	20
Sickle cell disease		
Yes	5	0.6
No	800	99.4
Cancer		
Yes	8	1.0
No	797	99.0

**Table 4 ijerph-19-11664-t004:** Factors associated with IgG seropositivity.

Variable	Total	IgG Seropositivity (n, %)	Bivariate			Multivariate		
			cOR	95%CI	*p*-Value	aOR	95%CI	*p*-Value
Age (years)	805	* 34 (IQR: 26–47)	0.994	0.985–1.00	0.260			
Sex								
Male	458	(223, 48.69)	1					
Female	347	(174, 50.14)	1.08	0.63–1.85	0.782			
District								
Magu	249	(112, 44.98)	1					
Misungwi	297	(135, 45.45)	1.01	0.73–1.42	0.912			
Ukerewe	259	(150, 57.92)	1.68	1.19–2.39	0.004	1.29	1.08–1.54	0.004
Occupation								
Business	145	(82, 56.55)	1.05	0.72–1.51	0.603			
Employed	56	(27, 48.21)	1.06	0.57–1.95	0.488	0.99	0.54–1.85	0.993
Farmer	448	(215, 47.99)	1.48	0.94–2.33	0.972			
Others	156	(73, 46.79)	1					
History of fever								
Yes	252	(127, 50.39)	1.16	0.79–1.43	0.679			
No	553	(270, 48.82)	1					
History of fatigue								
Yes	265	(133, 50.18)	1.05	0.56–1.12	0.193			
No	540	(264, 48.89)	1					
History of sore throat								
Yes	93	(45, 48.39)	0.96	0.62–1.48	0.849			
No	712	(352, 49.4)	1					
History of cough								
Yes	220	(97, 44.09)	0.75	0.55–1.02	0.069			
No	585	(300, 51.28)	1					
History of runny nose								
Yes	275	(139, 50.54)	1.08	0.81–1.44	0.616			
No	530	(258, 48.68)	1					
History of SOB								
Yes	52	(24, 46.15)	0.87	0.49–1.53	0.637			
No	753	(373, 49.52)	1					
History of chest pain								
Yes	32	(15, 46.88)	0.90	0.44–1.83	0.778			
No	773	(382, 49.42)	1					
History of smell loss								
Yes	64	(35, 54.69)	1.26	0.76–2.11	0.371	1.15	0.67–1.98	0.601
No	741	(362, 48.85)	1					
History of taste loss								
Yes	82	(45, 54.87)	1.28	0.81–2.03	0.289	1.27	0.78–2.06	0.331
No	723	(352, 48.69)	1					

* Median age of the IgG seropositive participants.

**Table 5 ijerph-19-11664-t005:** Factors associated with IgM positivity.

Variable	Total	IgM Seropositivity (n, %)	Bivariate Analysis			Multivariate Analysis		
			cOR	95%CI	*p*-Value	aOR	95%CI	*p*-Value
Age(years)	805	* 32.5 (IQR: 25–45)	0.994	0.976–1.01	0.594			
Sex								
Male	347	(26,7.52)	1.08	0.63–1.85	0.783	1.19	0.68–2.06	0.528
Female	458	(32,6.90)	1					
District								
Misungwi	297	(11, 3.70)	1					
Magu	249	(21, 8.43)	2.39	1.13–5.06	0.022	2.89	1.34–6.25	0.007
Ukerewe	259	(26, 10.03)	2.9	1.40–5.99	0.004	3.55	1.68–7.47	0.001
Occupation								
Business	145	(11, 7.58)	0.83	0.41–1.67	0.603			
Employed	56	(6, 10.71)	1.44	0.51–4.03	0.48			
Farmer	448	(29, 6.47)	0.99	0.42–2.31	0.97			
Others	156	(12,7.69)	1					
History of fever								
Yes	252	(20, 7.94)	1.6	0.67–2.05	0.588			
No	553	(38, 6.87)	1					
History of fatigue								
Yes	265	(24, 9.05)	1.48	0.86–2.55	0.157			
No	540	(34, 6.29)	1					
History of sore throat								
Yes	93	(6, 6.45)	0.88	0.37–2.09	0.765			
No	712	(52, 7.30)	1					
History of cough								
Yes	220	(15, 6.81)	0.92	0.50–1.69	0.795			
No	585	(43, 7.35)	1					
History of runny nose								
Yes	275	(28,10.18)	1.89	1.11–3.23	0.123	1.84	1.03–3.51	0.036
No	530	(30,5.67)	1					
History of SOB								
Yes	52	(6, 11.54)	1.76	0.72–4.31	0.217			
No	753	(52, 6.90)	1					
History of chest pain								
Yes	32	(1, 3.12)	0.41	0.05–3.02	0.378			
No	773	(57, 7.37)	1					
History of smell loss								
Yes	64	(6, 9.38)	1.37	0.56–3.33	0.486			
No	741	(52, 7.02)	1					
History of taste loss								
Yes	82	(12,14.63)	2.52	1.28–4.99	0.008	1.84	1.12–4.48	0.023
No	723	(46, 6.36)						

* Median age of the IgM seropositive participants.

## Data Availability

All data generated/analyzed during this study are included in this manuscript. Raw data can be obtained upon request from the director of Research and Innovation of the Catholic University of Health and Allied Sciences.

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
