# Peer review of "High Seroprevalence of SARS-CoV-2 in Mwanza, Northwestern Tanzania: A Population-Based Survey"

_ijerph, 2022, doi:10.3390/ijerph191811664_

Round 1
Author Response
|
Reviewers’ comments |
Response to the comments from the reviewer (Thanks for invaluable comments) |
|
The subjects included in the survey (805) are not sufficiently identified with respect to the population from which they were drawn, and it is not indicated how representative they are |
The description of the population has been given in the study area and study population. However representation in relation to total population has been outlined as one of the limitation of the study. |
|
The population size and features are not specified, either by district or by village. |
The population demographics have been added to the document |
|
Seropositivity: this point needs to be clarified by the authors as data in the text (rows 158-160) do not match what is presented in Table 2. |
The table has been omitted and the Venn diagram has been added |
|
The authors state an overall seropositivity of 49.8%, but upon evaluation based on the data provided it would be 50.4% |
The seropositivity has been recalculated from the Venn diagram which has been added |
|
As far as geographical distribution is concerned (rows 168-170), the authors hypothesize correlation with a road evidently of significant importance in the region. At the map level there is no evidence of this |
The statement has bee omitted from the draft just to maintain the distribution which can be appreciated |
|
It is also felt that Figures 4 and 5 may be redundant if they do not contribute more specific data to the understanding of the text |
The figures have been omitted from the draft |
|
Row 186 Table 6 doesn’t exist • • |
The error has been omitted |
|
Row 188-189 "living in Manu district was protective..... " The statement assumes that there should be some evidence to explain why the protective effect; more likely it is a lower prevalence of recent-onset infections, which if significant should be addressed in the discussion. |
The protective effect has been discussed |
|
as far as the discussion is concerned, it should probably be expanded and address the relatively low finding of IgM positivity in relation to the social situation of the population, individual protective measures possibly in use, or a feasible phase of epidemic reduction
|
Discussion has been expanded |

Reviewer 2 Report
COVID-19 pandemics is the major public health concerns worldwide. Nyawale and colleagues focus on the level and distribution of SARS-CoV-2 virus in Mwanza, Tanzania. This study was carried out by collecting blood samples from about 800 people and further detecting the IgM and IgG SARS-CoV-2 antibodies. Finally, they correlate the symptoms of SARS-CoV-2 infections and the serological results and then understand their relationship with the geological location of seropositive people. They found that about half of the samples are seropositive, and the symptoms of SARS-CoV-2 infections, especially running nose and loss of taste are associated with SARS-CoV-2 IgM seropositivity. This study is important in the local intervention control measures, but a number of major issues must be addressed.
Major comments:
1. 3 districts are selectively studied but the rationale of these choices is not presented clearly.
2. The results in paragraphs are not clearly presented or even missing in figures,
a. Line140: 55.7% (448) farmers. Where does it come from?
b. Line 144-146: This result is not clearly/ directly presented in Fig 1 but it requires readers to do some calculations to understand the data.
c. Line 152-153: Same as the above
d. Line 157: “The overall SARS-CoV-2 seropositivity was 49.8%....” Where does this value come from?
e. Line 163-164: “..the clustering of IgG seropositivity was observed in every district Fig 1.” IgG events were not found in Nyamagana. Also, this result is not found in Fig 1.
f. Line 169-170: Mwanza-Musoma Road is not clearly presented in figures.
3. Considering the statistical analysis (Table 3 and 4), bivariate and multivariate approaches are applied, but the authors do not justify which methods are appropriate for the analysis. Some data was selectively picked for both analysis while some was only analyzed with bivariate analysis. In addition, line 176 and line 184, univariate analysis is mentioned. Is it different from bivariate analysis?
4. Line 188: Table 6 is missing.
5. Line 198: What does the main road mean?
6. With all these errors, the conclusion is not supported by the results.
7. The difference in the IgG and IgM level has to be discussed. Are there any implications? References about the relationship between IgM seropositivity and the symptoms of SARS-CoV-2 infections have to be quoted.
8. Proof-reading has to be done carefully. There are lots of typo-errors.
Author Response
|
Reviewers comment |
Response to the reviewer (Thanks for invaluable comments) |
|
1. 1. 3 districts are selectively studied but the rationale of these choices is not presented clearly.
|
The rationale for the choice of the three districts have been outlined in the document.
|
|
2. a) Line140: 55.7% (448) farmers. Where does it come from? |
This has been well explained, that majority of the participants were involved in the farming activities
|
|
2. b) This result is not clearly/ directly presented in Fig 1 but it requires readers to do some calculations to understand the data.
|
A table instead of figure has been added |
|
2.c) c. Line 152-153 This result is not clearly/ directly presented in Fig 1 but it requires readers to do some calculations to understand the data. |
A table instead of a figure has been added |
|
2.d) “The overall SARS-CoV-2 seropositivity was 49.8%....” Where does this value come from? |
The overall seropositivity was recalculated from summation of each IgG and IgM positivity. The previous table has been omitted and a different figure(Venn diagram has been added for easy understanding) |
|
2.e Line 163-164: “..the clustering of IgG seropositivity was observed in every district Fig 1.” IgG events were not found in Nyamagana. Also, this result is not found in Fig 1 |
The statement has been modified into the “IgG positivity was observed in the three districts of Magu, Misungwi and Ukerewe” |
|
2.f Line 169-170: Mwanza-Musoma Road is not clearly presented in figures |
Mwanza Musoma road has been omitted from the draft |
|
Considering the statistical analysis (Table 3 and 4), bivariate and multivariate approaches are applied, but the authors do not justify which methods are appropriate for the analysis. Some data was selectively picked for both analysis while some was only analyzed with bivariate analysis. In addition, line 176 and line 184, univariate analysis is mentioned. Is it different from bivariate analysis |
The tables have rechanged to table 4 and 5
The analysis into factors associate was done using bivariate analysis first and the factors with p value less than 0.5 were further analyzed into multivariate analysis, how-ever, the variable with p value less than 0.005 was considered statically significant
Line 176 and 184, the changes on the draft have been made to fit the context of what was done on the analysis. |
|
4. Line 188: Table 6 is missing.
|
The error has been omitted |
|
5. Line 198: What does the main road mean |
The statement has been omitted from the document |
|
Proof-reading has to be done carefully. There are lots of typo-errors |
Proof reading was extensively with assistance of the native English speaker |

Round 2
Reviewer 2 Report
The revised version is highly improved, but a number of errors or concerns must be addressed.
Major comments:
1. Line 154: Most of the participants were farmers so what is the relationship with your results? Why do you emphasize this nature?
2. Two mainland area and one island area are selected in this study. However, the relationship between the geographical factor and the seropositivity is not clearly discussed. Nyawale and colleagues tried to correlate the serological difference found in Misungwi and Magu with their accessibility. However, the number of Ig+ events in Ukerewe is the highest given that Ukerewe is an island with limited connection to other areas. Moreover, there was few discussion about the findings in Ukerewe.
3. Reference 18 is not appropriately quoted because it did not show the relationship between IgM seropositivity and the history of runny nose.
4. Line 249-250: The sentence is not complete.
5. It is highly recommended to explain the reason to test the IgG and IgM level in blood, which helps reader to understand your discussion.
6. There are so many versions of ‘SARS-CoV-2’ presentation along the whole manuscript. The presentation must be unified.
7. Reference is still missing on line 124.
8. Typo-/grammatical errors have been improved but there are still some, eg, line 38, 39, 65, 71, 117, 242, 246, etc.
9. Wordings should be used more precisely. For example, in the discussion section, ‘observed’ could be replaced with ‘revealed’, ‘showed’, ‘indicated’, etc.
1. Line 38-39: ‘Coronavirus’ is a single word, and the disease is caused by the virus not pandemic.
Author Response
|
REVIEWER’ COMMENTS |
RESPONSE TO THE REVIEWER |
|
1. Line 154: Most of the participants were farmers so what is the relationship with your results? Why do you emphasize this nature? |
Thank you. The sentence has been omitted in this draft |
|
2. Two mainland area and one island area are selected in this study. However, the relationship between the geographical factor and the seropositivity is not clearly discussed. Nyawale and colleagues tried to correlate the serological difference found in Misungwi and Magu with their accessibility. However, the number of Ig+ events in Ukerewe is the highest given that Ukerewe is an island with limited connection to other areas. Moreover, there was few discussions about the findings in Ukerewe. |
Thank you. We have re-done analysis, residing in Ukerewe district was significantly associated with IgG seropositivity. The observation is right and we have discussed the seropositivity findings in Ukerewe in this draft. Considering Ukerewe as an Island with a lot of fishing activities, the interactions might be higher than in Magu and Misungwi. |
|
3. Reference 18 is not appropriately quoted because it did not show the relationship between IgM seropositivity and the history of runny nose |
Thank you. The reference on that particular sentence has been omitted and citations have revised |
|
4. Line 249-250: The sentence is not complete. |
Thank you. The sentence has been correctly revised and corrected |
|
5. It is highly recommended to explain the reason to test the IgG and IgM level in blood, which helps reader to understand your discussion |
Thank you. The study aims at establishing seroprevalence therefore the right sample is blood to establish IgG and IgM as markers. The use of blood as a specimen of choice for this study has been explained |
|
6. There are so many versions of ‘SARS-CoV-2’ presentation along the whole manuscript. The presentation must be unified |
Thank you. SARS-Cov-2 has been used consistently in this version |
|
7.Reference is still missing on line 124 |
Thank you. The extraction of the sera has been elaborated |
|
8. Typo-/grammatical errors have been improved but there are still some, eg, line 38, 39, 65, 71, 117, 242, 246, etc |
Thank you. The typo-/grammatical errors have been reviewed. |
|
9. Wordings should be used more precisely. For example, in the discussion section, ‘observed’ could be replaced with ‘revealed’, ‘showed’, ‘indicated’, etc |
Thank you. The wording has been worked on in this draft. |
|
10. Line 38-39: ‘Coronavirus’ is a single word, and the disease is caused by the virus not pandemic. |
Thank you. The sentence has been revised |
